# Skin Signals: Exploring the Intersection of Cancer Predisposition Syndromes and Dermatological Manifestations

**DOI:** 10.3390/ijms26136140

**Published:** 2025-06-26

**Authors:** Ilse Gabriela Ochoa-Mellado, Alejandra Padua-Bracho, Paula Cabrera-Galeana, Rosa María Alvarez-Gómez

**Affiliations:** 1Clínica de Cáncer Hereditario, Instituto Nacional de Cancerología, Mexico City 14080, Mexico; doctoragen.ilse@gmail.com (I.G.O.-M.); alejandra.padua@gmail.com (A.P.-B.); 2Subdirección Oncología, Instituto Nacional de Cancerología, Mexico City 14080, Mexico; drapaulacabrera@gmail.com

**Keywords:** hereditary cancer, skin, early diagnosis, dermatological manifestations, genodermatosis

## Abstract

Cutaneous manifestations can serve as early and sometimes the first clinical indicators in various hereditary cancer predisposition syndromes. This review provides a comprehensive overview of the dermatological signs associated with these syndromes, aiming to facilitate their recognition in clinical practice. Hereditary Breast and Ovarian Cancer syndrome is notably linked to an increased risk of melanoma. BAP1 tumor predisposition syndrome is characterized by BAP1-inactivated melanocytic tumors. Muir–Torre syndrome, a variant of Lynch syndrome, presents with distinctive cutaneous neoplasms such as sebaceous carcinomas, sebaceous adenomas, and keratoacanthomas. PTEN hamartoma tumor syndrome commonly features hamartomatous growths, trichilemmomas, acral keratoses, oral papillomas, and genital lentiginosis. Gorlin syndrome is marked by basal cell carcinomas and palmoplantar pits, while Peutz–Jeghers syndrome is identified by mucocutaneous pigmentation. In familial adenomatous polyposis, the cutaneous findings include epidermoid cysts, fibromas, desmoid tumors, and lipomas. Additionally, we examined monogenic disorders associated with cancer risk and skin involvement, such as xeroderma pigmentosum, neurofibromatosis type 1, familial atypical multiple-mole melanoma syndrome, and Fanconi anemia. The early recognition of these dermatologic features is essential for a timely diagnosis and the implementation of appropriate surveillance strategies in individuals with hereditary cancer syndromes.

## 1. Introduction

Hereditary cancer predisposition syndromes account for approximately 10% of all cancer etiologies [1]. These syndromes are variable and present with distinctive tumors and/or clinical features that help us to suspect and determine the diagnosis [2]. In some cases, cutaneous manifestations could be the first sign that we observe, for example, mucocutaneous pigmentations seen in Peutz–Jeghers syndrome [3]; palmar pits in Gorlin syndrome [4]; nonmalignant neoplasms, presenting as sebaceous adenomas or epitheliomas, seen in Muir–Torre syndrome [5]; or even malignant neoplasms such as basocellular carcinoma (BCC), squamous cell carcinoma (SCC), and cutaneous melanoma.

This review aims to provide a comprehensive overview of all the dermatologic manifestations observed in both hereditary cancer syndromes and monogenic diseases. The goal was to highlight the key clinical signs that may be detected during a routine examination and emphasize the importance of close collaboration among dermatologists, oncologists, medical geneticists, and other specialists in hereditary cancer. By fostering interdisciplinary coordination, this approach ensures thorough patient management, from the initial suspicion to confirmatory testing and long-term follow-up.

## 2. Hereditary Cancer Syndromes and Skin Manifestations

Hereditary cancer syndromes are caused by germline pathogenic variants (PVs) that significantly increase an individual’s risk of developing certain types of cancer, depending on the gene affected. Syndromes such as Hereditary Breast and Ovarian cancer (HBOC) syndrome, BAP1 tumor predisposition syndrome, Lynch syndrome (LS), PTEN hamartoma tumor syndrome (PHTS), Bannayan–Riley–Ruvalcaba syndrome (BRRS), Cowden syndrome (CS), Proteus syndrome (PS), Gorlin syndrome (GS), Peutz–Jeghers (PJ) syndrome, and familial adenomatous polyposis (FAP) often exhibit distinctive dermatological features that can serve as early clinical indicators of an underlying genetic predisposition to malignancy. A comprehensive overview of these hereditary cancer syndromes and their corresponding skin manifestations are presented in this section, as well as summarized in Figure 1 and Table 1.

### 2.1. Hereditary Breast and Ovarian Cancer Syndrome and Melanoma

HBOC syndrome is associated with pathogenic or likely pathogenic variants (P/LP), mainly in the tumor suppressor BRCA1/2 genes, with a prevalence of 1 to 400 individuals. P/LP variant carriers have an increased risk of breast, ovarian, prostate, pancreatic, and melanoma cancer [6]. Regarding skin cancer or melanoma, it has not been clearly defined whether these neoplasms are manifestations of the BRCA1/2 mutation carrier status [7,8,9]. Specifically for melanoma, several cohort studies are trying to determine the association with variants of BRCA2, with inconsistent results; some have shown an increased risk [10,11]. In addition, some authors have pointed out that the etiology of melanoma is complicated due to its polygenic factors, such as nevi and pigmentation [12]. There are no established guidelines for skin cancer screening in BRCA1/2 carriers; however, the NCCN guidelines propose skin exploration annually for patients with pathogenic variants in the BRCA2 gene [13].

### 2.2. BAP1 Tumor Predisposition Syndrome

BRCA1-associated protein 1 (BAP1), encoded by the tumor suppressor gene BAP1, has de-ubiquitinating activity to regulate proteins in DNA damage repair pathways. Pathogenic variants in the BAP1 gene increase the risk of tumor predisposition to uveal and cutaneous melanoma, malignant mesothelioma, and renal cell carcinoma [14,15]. The prevalence of this syndrome remains unknown. The manifestations associated with melanoma are called BAP1-inactivated melanocytic tumors (BIMT); clinically, there are multiple (two or more) dome-shaped papules with a central pink, nonstructural area and pigmented globules on dermoscopy [16]. The main sites are the head and neck, the trunk, and the extremities. The age of onset is usually in the second and third decades of life [17]. A histopathological diagnosis is challenging, with a differential diagnosis ranging from benign Spitz nevus to nevoid melanoma. Therefore, a comprehensive analysis, including immunohistochemistry (IHC) and a molecular analysis, is required. Another characteristic and defining signal is the presence of the BRAFV600E variant, which is rarely observed in Spitz nevi [17].

The penetrance of BAP1-inactivated melanocytic tumors is 75% to 100%. It has been reported that individuals with multiple BIMTs or a personal history of BAP1-related cancers have a 70% chance of having a pathogenic variant of *BAP1*. Multiple BCCs are also present in patients with this tumor predisposition syndrome, although we need to consider that this is a common cancer in the population; however, for those with multiple types of cancer or early-stage cancer, it is important to evaluate and make a differential diagnosis with Gorlin syndrome [18].

Some recommendations for surveillance are a dermatologic examination annually starting at 18–20 years old and a self-skin examination according to the ABCDE characteristics of melanoma; alternatively, whole-body imaging of the skin annually may be an option [19].

### 2.3. Lynch Syndrome

Lynch syndrome (LS) or non-polyposis colorectal cancer syndrome is an autosomal dominant predisposition cancer syndrome characterized by the presence of colorectal cancer and other extracolonic malignancies like endometrial, ovarian, stomach, small bowel, pancreatic, bile duct, upper urinary tract, and cutaneous tumors. The latter are present in 3–5% of extracolonic manifestations [5]. This syndrome is caused by pathogenic germline variants in the mismatch repair genes MLH1, MSH2, MSH6, and PMS2 and a deletion in the 3′ end of the EPCAM gene, leading to the hypermethylation and transcriptional silencing of MSH2. The prevalence is estimated at between 1 in 2000 individuals.

Skin neoplasms, including sebaceous carcinomas, sebaceous adenomas, and keratoacanthomas, develop in some individuals with LS [20]. This variant of LS is known as Muir–Torre syndrome (MTS), and is seen in 9.2% of patients with LS [21]. The literature refers to an association of pathogenic variants of the MSH2 gene in over 90% of MTS patients, followed by the MLH1 gene, while just a few reports have associated MTS with MSH6 and PMS2. Skin neoplasms are also significantly associated with the male sex, an older age, and Caucasian men [22].

Skin manifestations could be the only neoplasm or the earliest manifestation of LS in approximately 30% of patients [22].

A cohort of 581 patients with MTS exhibited sebaceous adenomas in 43%, sebaceous carcinoma in 27%, keratoachantomas in 16%, sebaceous epithelioma in 13%, and BCC in 10% of the cases [5].

Sebaceous carcinoma is an adenocarcinoma with varying degrees of sebaceous differentiation. It tends to appear on the eyelids, but can be classified as a periocular or extraocular tumor [23]. Periocular tumors arise from the tarsal glands on the upper eyelid. The appearance of the lesion is a papule, nodule, or cystic lesion with a yellow-pink color, firm, painless, and rapidly enlarging; we can even find the lesion with ulceration. Extraocular tumors are present on the head or neck, and in rare cases, on the foot, penis, or vulva; clinically, the tumor is a yellow to pink or red nodule, and very unspecific. A diagnosis is made with the help of histopathology and the management of the lesions involves their complete excision, but periocular tumors need an evaluation for reconstructive techniques due to their location [21].

Sebaceous adenomas, sebaceomas, and sebaceous epitheliomas are benign neoplasms with sebaceous differentiation. Some are asymptomatic and appear in areas with rich sebaceous glands, such as the head and neck. Clinically, they are multiple or solitary nodules with a yellow-red color, which can be multilobulated. The difference between these two lesions is the presence of basaloid cells on histopathology; sebaceomas have more than 50% basaloid tumor cells, while sebaceous adenomas have less than 50% basaloid cells [21]. The treatment for these lesions is an excisional biopsy.

A keratoacanthoma is a rapidly growing, dome-shaped skin tumor with a centralized keratinous plug. They originate from the pilosebaceous unit due to the hyperkeratosis of the infundibulum and are more common in hair-bearing and sun-exposed zones. These tumors have a period of rapid growth followed by establishment and, finally, involution. Although they are considered benign tumors, their treatment is an excisional biopsy or laser treatment, depending on the size of the lesion. In a cohort of 581 patients with MTS, keratoacanthomas were found on 16% of the patients [5].

BCCs and SCCs are also present in patients with LS or MTS, more frequently than sebaceous neoplasms. However, we need to consider multiple factors, such as sun exposure, that can increase the incidence of these lesions in patients with a predisposition syndrome.

The “Mayo MTS risk score” helps to identify patients who are more likely to have MTS and who should undergo a molecular test. It includes the age at the presentation of the lesion, the total number of sebaceous neoplasms, the personal history of Lynch-related cancer, and the familial history of Lynch-related cancer. The score ranges from 0 to 5, and a score of 2 or more has a sensitivity of 100% and a specificity of 81% for predicting a pathogenic germline variant in a patient with LS [21].

Once diagnosed, patients require an integral evaluation and surveillance. The literature recommends a full-body dermatological evaluation for patients with LS or MTS, as we now know that neoplasms on the skin can not only be on the face, but also on the trunk or low on the face and neck in 25% of cases [22,24].

### 2.4. PTEN Hamartoma Tumor Syndrome

PTEN hamartoma tumor syndrome (PHTS) is recognized as a spectrum of disorders caused by pathogenic variants of the tumor suppressor gene PTEN, including Cowden syndrome (CS) and Bannayan–Riley–Ruvalcaba syndrome (BRRS), which are caused by germline variants, and PTEN-related Proteus syndrome (PS) and Proteus-like syndrome, which are caused by mosaic variants [25]. Cowden syndrome is the most known and recognizable clinical syndrome within the spectrum, and the estimated prevalence is 1 in 200,000 [26].

#### 2.4.1. Cowden Syndrome (CS)

Cowden syndrome is characterized by a predisposition to multiple hamartomas on any organ and cancer risks [26]. It is said that 99% of individuals with CS develop mucocutaneous manifestations by their third decade of life. It is an autosomal dominant condition, but 45% of cases arise de novo. There is an elevated risk of breast, thyroid, endometrial, colorectal, and melanoma cancers [26].

The skin manifestations of this syndrome are variable and include trichilemmomas, acral keratoses, and oral papillomas [26,27].

Trichilemmomas, which are present in 6–38% of CS cases [27], are skin-colored papules that primarily appear on the face and neck. The solitary lesion is not related to CS; therefore, patients have multiple trichilemmomas. Some histological findings include the folliculocentric lobular proliferation of polygonal, clear, PAS-positive isthmic cells with the nuclear palisading of the peripheral cells [27].

Acral keratoses, which are wart-like papules seen on the dorsum of the hands and feet, and are present in approximately 10% to 80% of CS cases [27]. These papules have a translucent aspect with or without a central pit. The histological findings are not specific, and show only orthokeratosis, hypergranulosis, and acanthosis [27].

Oral papillomas are papules that can coalesce, taking on a cobblestone appearance, and are seen on the lips, buccal mucosa, and gingiva, with a low frequency in the pharynx or larynx [28]. The frequency of these lesions in patients with CS is approximately 15% to 85%. Histologically, they comprise a fibrovascular core covered with a benign epithelium [27].

Mucocutaneus neuromas are hamartomas of the nerve sheath, and are reported with a frequency of 5% to 10% in patients with CS. However, these lesions are more common in patients with MEN2B variants [29]. These lesions have a predilection to appear on the extremities and face and are dome-shaped, translucent, and skin-colored papules that are painful. The histological findings are similar to those seen in MEN2B patients, but with a compact arrangement of well-delineated hypertrophic nerve bundles surrounded by a distinct perineural sheath [29].

Penile pigmentation is part of PHTS and is a major criteria for CS. It has a prevalence of 19–54% of cases, with another report finding a frequency of 40% in a cohort of patients aged 1 to 26 years [27]. It is recognized as hyperpigmented macules; histological findings show the hyperplasia of the epidermis with increased pigmentation in the basal layer and an increase in melanocytes.

Other lesions on the skin that are related to CS are lipomas, with a frequency of approximately 30–40%, in contrast to 15% within the general population [27]. Vascular malformations are seen in 18% to 34% of patients with CS, versus 5% to 10% in the general population [26]. This last type of lesion can be multifocal and intramuscular with high flow, and image studies are the best method for conducting an evaluation.

#### 2.4.2. Bannayan–Riley–Ruvalcaba Syndrome (BRRS)

BRRS is recognized as an autosomal dominant condition with a triad of macrocephaly, hyperpigmented penile macules, and hamartomatous tumors (lipomas, hemangiomas, and gastrointestinal polyps) [30]. Other clinical manifestations include intellectual disability, development delays, and skeletal manifestations such as pectum excavatum and scoliosis [31].

There are no specific characteristics on the skin; genitalia lentiginosis can be observed as café-au-lait macules (CALMs) with different sizes, but in the presence of other criteria like lipomas and vascular malformations, a suspicious presentation needs to be confirmed [30,32].

The risk of cancer is thought to be the same as that of CS, but no studies have been performed [33].

#### 2.4.3. Proteus Syndrome and Proteus-like Syndrome

Proteus syndrome (PS) is characterized by the hamartomatous overgrowth of multiple tissues, connective tissue nevi, epidermal nevi, and hyperostoses. These patients present germline pathogenic variants of PTEN, but the lesions on the affected organs are developed by a second hit [34].

Proteus-like syndrome refers to individuals with features of PS who do not meet the diagnostic criteria [33].

### 2.5. Gorlin Syndrome or Nevoid Basal Cell Carcinoma Syndrome

Gorlin syndrome (GS) is an autosomal dominant disorder characterized by various developmental abnormalities, including odontogenic keratocysts and BBCs, with the prevalence estimated to be between 1 in 57,000 and 1 in 164,000 individuals. It is caused by pathogenic variants of the membrane-bound protein encoded by the PTCH1 gene in 90% of patients [35]. Other genes associated with GS include SUFU and PTCH2. The SUFU gene is a part of a corepressor complex that provides additional negative regulation of GLI factors through direct binding, and is related to a high risk of medulloblastoma. PTCH2 (which encodes a protein that may function as a tumor suppressor in the hedgehog signaling pathway) is related to a milder phenotype than that of patients with PTCH1 [35].

The classic phenotypes in patients with GS are the presence of more than two BCCs or an age under 20 years; odontogenic keratocysts of the jaw; palmar or plantar pits; bilaminar calcification of the falx cerebri; bifid, fussed, or splayed ribs; macrocephaly; congenital malformations such as a cleft lip or palate; frontal bossing; a coarse face; and severe hypertelorism. Other skeletal abnormalities include a Sprengel deformity, a pectus deformity, or syndactyly of the digits [4,35].

In 80% of skin cancer cases, the initial symptoms are small black spots that are often misidentified as moles. Over time, these spots gradually increase in size. They usually appear on the mid-face, around the nose and upper lip, and do not cause pain or itching [35]. Another study reported that, apart from the face, the topography includes non-sun exposure zones such as the upper back and the upper extremities [4]. Histologically, this type of cancer is indistinguishable from sporadic BCC [4].

Palmoplantar pits are present in 80–90% of patients and are the most prevalent feature of GS; they are nonpalpable, shallow punctiform depressions of 2–3 mm in diameter. Histological observations reveal an abrupt zone or hypokeratosis, with variable hypogranulosis and parakeratosis and basal cells with hyperplasia. The typical topography is more frequently observed on the palmar surfaces than on the plantar surfaces, specifically on the thenar and hypothenar eminences and the arches of the feet. Once these manifestations appear, they are permanent [4].

Other skin findings include multiple melanocytic nevi, which are present in 50% of cases and can increase with age. Milia can be found in 60% of cases on the periorbital and malar regions, and the nose. Multiple epidermal cysts can also be found in 50% of patients [36].

Vismodegib is a newly approved target therapy for advanced BCCs. It is an effective treatment not only for BCCs, but also for odontogenic keratocysts [36].

Management also consists of avoiding excessive UV exposure and using protective measures against the sun [37].

### 2.6. Peutz–Jeghers Syndrome

Peutz–Jeghers syndrome (PJS) is an autosomal dominant genodermatosis caused by germline pathogenic variants of STK11, a tumor suppressor gene. The syndrome is characterized by gastrointestinal hamartomatous polyps, hyperpigmentation of the skin and mucosa, and an increased risk of developing tumors such as ovarian/testicular, pancreatic, breast, and uterine neoplasia [38]. It has an incidence of 1 in 25,000 to 1 in 280,000 individuals [3].

On the skin and mucous membranes, the first manifestation is hyperpigmentation, even before the gastrointestinal polyps appear. There are oval or round macules with a size of 2 to 5 mm and a dark brown to black color. The topography is on the face around the eyes, mouth, fingers, and feet; around orifices; and on the mucous membranes of the cheeks [39].

The presence of skin and mucocutaneous macules around the orifices and within the oral cavity, combined with a history of gastrointestinal bleeding, is indicative of PJS. Therefore, it is mandatory to conduct a gastrointestinal endoscopy or colonoscopy, accompanied by a complete examination by a geneticist and an assessment of the familial history [40].

### 2.7. Familial Adenomatous Polyposis

Familial adenomatous polyposis (FAP) is an autosomal dominant disease characterized by the presence of hundreds to thousands of adenomatous polyps in the gastrointestinal mucosa, along with distinct extraintestinal manifestations. The incidence of this condition ranges from 1 in 7000 to 1 in 30,000 births [41]. It is associated with variants in the APC gene, a tumor suppressor that regulates intracellular B-catenin levels. Clinical manifestations include desmoid tumors, small intestinal adenomas, and carcinomas (such as colorectal), as well as others manifestations like osteomas, dental anomalies, skin lesions, and congenital hypertrophy of the retinal pigment epithelium.

The following skin lesions may be present in this syndrome: fibromas, epithelial cysts, desmoid tumors, lipomas, leiomyomas, neurofibromas, and pilomatrixomas. The latter three are less common [42]. The primary lesion is an epithelial cyst, which is observed in 50% to 100% of cases, with a tendency to appear on the scalp, face, and neck [43]. Histologically, the cysts are characterized by a lesion on the dermis that is filled with keratin. The wall of the cyst is composed of squamous stratified epithelium. In addition, Gardner-associated fibromas are of particular relevance. Recognized in the 2013 WHO classification, this benign soft tissue lesion is associated with FAP [41,44]. The lesions primarily occur on the back and paraspinal regions, but may also be found on the head, neck, and extremities. A histopathological examination reveals distinctive features, including dense thick bands of collagen interspersed with small bland fibroblasts arranged in random bundles. Notably, slender clef-like spaces can be observed between the collagen bundles, imparting a “cracked” appearance [41,44]. Furthermore, desmoid tumors represent fibroblastic soft tissue tumors with an intermediate malignant potential. These tumors exhibit local invasiveness and destructiveness, but never metastasize. They are present in approximately 10 to 25% of individuals with familial adenomatous polyposis [45]. Common locations for these tumors include the abdominal wall, the intra-abdominal area (mesenteric fibromatosis), and extra-abdominal areas, including the head, neck, and extremities (fibromatosis) [41]. The tendency for intra-abdominal tumors to predominate and the stimulating effect of surgery are among the distinguishing features that set FAP-associated desmoid disease apart from sporadic cases [46]. The histopathological features on intra-abdominal tumors include the presence of a solid, firm, white mass with a diffuse growth of uniform, elongated spindle cells within a collagenous myxoid stroma, characterized by bundles of collagen with a glassy appearance. In the case of extra-abdominal fibromatosis, the presentation may include the infiltration of local structures and skeletal muscle fibers, with scattered clusters of atrophic skeletal muscles observed. The fibromatosis typically exhibits bland spindle cells with long fusiform nuclei and metachromatic matrix material [41].

In a study on the prevalence of skin lesions in patients with FAP conducted by Burger et al., lipomas were identified as the most common lesion observed [42]. However, lipomas are the most common soft tissue tumors in the general population, which may explain why they are more representative in the study cohort. The epidemiological data are limited, since lipomas typically do not cause discomfort or require medical attention, leading them to often go undetected. Further studies are still needed to determine the difference in the prevalence of lipomas among patients with FAP.

**Table 1 ijms-26-06140-t001:** Hereditary cancer syndromes with skin manifestations.

Syndrome	Diagnostic Criteria	Dermatologic Manifestations	References
HBOC syndrome and melanoma	NCCN guidelines^®^ [13]	Melanoma	[12]
*BAP1* tumor predisposition syndrome	Rai et al. 2016 [19]	BAP1-inactiveted melanocytic tumors; basal cell carcinoma	[17]
Lynch syndrome	NCCN guidelines^®^ [47]	Sebaceous carcinomas, adenomas, and keratoacanthomas; BCC and SCC	[5]
*PTEN* hamartoma tumor syndrome	Pilarski et al. (2013) [48]	Hyperpigmented penile macules, hamartomatous tumors, thrichilemmomas, acral keratoses, papillomas, epidermal nevi, hyperostoses	[27,30,47]
Nevoid basal cell carcinoma syndrome	Fernandez Lt et al. (2022) [4]	BCC; palmar or plantar pits; melanocytic nevi; milia; multiple epidermal cysts	[49]
Peutz–Jeghers syndrome	NCCN guidelines^®^ [47]	Hyperpigmentation of the skin and mucous membranes	[39]
Familial adenomatous polyposis	NCCN guidelines^®^ [47]	Fibromas, epithelial cysts, desmoid tumors, lipomas, leiomyomas, neurofibromas, and pilomatrixomas	[42,43]

## 3. Monogenetic Disease with Cancer Predisposition and Skin Manifestations

In contrast to hereditary cancer syndromes, other genetic conditions—including genodermatoses, a group of rare genetic disorders primarily affecting the skin; monogenic diseases, caused by mutations in a single gene; and other genetic syndromes—can also increase the risk of certain types of cancer while presenting diverse skin manifestations. The following section provides a comprehensive overview of these conditions, including genodermatosis (xeroderma pigmentosum, neurofibromatosis type 1, familial atypical multiple-mole melanoma syndrome, and Rothmund-Thomson syndrome), monogenic diseases (Bloom syndrome and Wermer syndrome), and other genetic syndromes (Fanconi anemia and dyskeratosis congenita). A summary is presented in Table 2 and Figure 2.

### 3.1. Xeroderma Pigmentosum

Xeroderma pigmetosum (XP) is considered a genodermatosis with an autosomal recessive inheritance pattern, occurring in 250,000 live births. This condition is characterized by “dry and pigmented skin”, which is how Kaposi coined the name of the syndrome. The etiology of XP is a germline pathogenic variant of genes involved in the nucleotide excision repair (NER) pathway. These genes include XPA, XPB, XPC, XPD, XPE, XPF, XPG, and POLH. Depending on which gene is affected by the pathogenic variant, XP can present with different complementation groups, illustrating the locus heterogeneity of the syndrome [50].

Clinical manifestations include photosensitivity, early-onset ultraviolet radiation (UVR)-induced skin pigmentary changes, UVR-induced damage to the eyes, an increased risk of cutaneous tumor development, and, in some cases, neurological degeneration [50].

The skin undergoes significant changes throughout life as a result of the increased photosensitivity. Although the skin generally presents as normal at birth, it becomes damaged over time. The subtypes of XP associated with severe and exaggerated sunburns, which may even result in blistering with minimal sun exposure, include XPA, XPB, XPD, XPF, and XPG. Conversely, XPC, XPE, and other XP variants exhibit less severe sunburns, but still manifest with abnormal skin pigmentation, including lentigos and freckles [50].

The consequences on the skin over time are premature aging, progressive xerosis, atrophy, wrinkling, telangiectasia, early-onset lentiginosis, and poikiloderma. Also, the presence of hypopigmented areas, known as a “salt and pepper appearance”, is due to melanocytes that have lost their ability to produce melanin [50].

The skin cancers that develop in patients with XP, in order of prevalence, are SCC, BCC, and malignant melanoma. Individuals at risk for non-melanoma skin cancers experience a 10,000-fold increased risk, while the risk for malignant melanoma is heightened by 2000-fold [50,51]. The age at the time of presentation for non-melanoma skin cancers is 9 years, whereas melanoma appears at 22 years old. Patients with XP also have an increased frequency of other skin cancers, including keratoacanthoma, epithelioma, sebaceous cell carcinoma, fibrosarcoma, and angiosarcoma [50]. In addition to skin cancer, patients develop internal cancers such as gliomas, schwannomas, astrocytoma, gastric cancer, and lung cancer [52]. Other studies have noted an elevated risk of developing breast, thyroid, uterine, hematological, and kidney malignancies [53].

The histological findings in patients with XP include atrophy, hyperkeratosis, and epidermal depressions filled with keratin fibers; there is also an impairment of immune cell infiltrates into the dermis, and irregular areas of abundant melanin-containing pigmented cells. Additionally, there is an increased number of melanocytes in the basal lamina and typical dysplasia, with the latter being recognized as the main feature observed [54].

The management of these patients involves the use of physical photoprotectors such as zinc oxide, red ferric oxide, talc, and nanoparticles, as well as chemical lotions with aminobenzoates that selectively absorb UVB light and benzophenones, which absorb UVA light. Sunscreen should be applied every 2 h. Additionally, oral retinoids, such as acitretin and isotretinoin, have shown protective effects against BCCs and SCC [54].

### 3.2. Neurofibromatosis Type 1

Neurofibromatosis type 1 (NF1), with autosomal dominant inheritance, is classified as a multisystem disease. It has an incidence of 1 in 3000 individuals and is caused by a germline pathogenic or likely pathogenic variant of the NF1 gene. This gene encodes neurofibromin, a ubiquitous protein that is mainly expressed in neurons, Schwann cells, and glial cells.

The most common clinical manifestations include café-au-lait spots (CALMs), ephelides, and neurofibromas [55]. Patients with NF1 have a cancer incidence that is five to ten times greater than in the general population, with 40% of patients developing cancer by the sixth decade of their life [36]. The most frequently observed malignant tumors are low-grade gliomas and malignant peripheral nerve sheath tumors. Women with NF1 have an up to 5-fold increased risk of breast cancer before the age of 50, and an overall 3.5-fold increased risk of breast cancer [56].

CALMs appear on almost 100% of the patients with NF1. They typically appear as flat, hyperpigmented macules with defined and regular borders [57], often manifesting in a generalized and dispersed form on the skin. The freckles are small, pigmented lesions of a light brown color that tend to appear in areas with minimal sun exposure, such as folds (the armpits, the inguinal region, the skin above the breast, the base of the neck, and the upper eyelids), and appear on >90% of the patients [57]. The hallmark tumors in NF1 are neurofibromas, which are classified as cutaneous, subcutaneous, or plexiform. These benign tumors, arising from the peripheral nerve sheath, occur in 99% of patients (Schawn cells, mast cells, fibroblast axons, extracellular matrix, and perineural cells). Plexiform neurofibromas are observed in 20–30% of patients with NF1 [57].

The recommended surveillance is a complete examination of the skin annually.

### 3.3. Familial Atypical Multiple-Mole Melanoma Syndrome

Familial atypical multiple-mole melanoma (FAMM) syndrome is a dominant autosomal condition that is clinically characterized by numerous nevi (>50) and a family history of melanoma. In addition, the patients can present with pancreatic cancer or other cancers such as breast, lung, or nasopharyngeal SNC tumors [58,59]. This syndrome is associated with CDKN2A, a tumor suppressor gene that encodes two proteins, p16 and p14, both of which regulate the cell cycle. The prevalence is unknown due to a variable expressivity, and the penetrance varies for geographical areas, estimated at 30–91%, 50–76%, and 13–58% among patients aged 50–80 years in Australia, the United States, and Europe, respectively [58].

The clinical phenotype consists of numerous nevi, some of which are atypical (asymmetrical with color variability, border irregularity, and variable sizes) interspersed between benign-looking nevi [60].

These patients have an increased risk of melanoma at an early stage of disease onset. The patients seem to be more prone to developing superficial spreading and nodular melanomas with histological features such as a higher level of pigmentation and increased pagetoid scatter; however, they are significantly less invasive (lower Clark levels) and tend to appear on the trunk more than on the head and neck [58].

The surveillance of carriers of a pathogenic or likely pathogenic variant of CDKN2A is an exhaustive skin examination, including an examination of the scalp and oral and genital mucosa every 6 months with the help of dermoscopy and clinical photography. The recommendation is to avoid smoking after the description of an increased prevalence of tobacco-associated cancers in FAMM syndrome patients [59].

### 3.4. Fanconi Anemia

Fanconi anemia (FA) is an autosomal recessive genetic syndrome, with rare cases having X-linked or dominant negative inheritance. FA syndrome is characterized by a deficiency in DNA damage repair, which results in bone marrow failure and an increased risk for various epithelial tumors, including SCC of the head and neck and esophageal, anogenital tract, and skin cancers [61]. It affects 1 in 160,000 individuals, with a life expectancy of 20 years. It is caused by germline pathogenic or likely pathogenic variants that cause the loss of function of more than 20 genes (FANCA, FANCB, FANCC, FANCE, FANCF, FANCG, FANCL, FANCM, FANCT, FANCD2, FANCI, FANCJ, FANCN, FANCO, FANCP, FANCQ, FANCR, FANCS, FANCU, FANCV, and FANCW) [61].

The skin manifestations seen in FA syndrome are associated with altered pigmentations, such as café-au-lait macules, flexural hyperpigmentation, and hypopigmented macules, which are found in 68% of FA patients [62]. The CALMs have the particularity of being subtly darker than the surrounding skin, and are termed “shadow spots.” Hypopigmented macules are more frequent than those in the general population. Skin-fold freckle-like macules are more frequently hypopigmented and localized in clusters within the skin folds [62].

Usually, they can present as warts associated with a decrease in immunity because they are related to the human papilloma virus infection [63].

There is an association between FA and skin cancers in relation to the damage caused by ultraviolet rays to the DNA; these patients do not have the ability to repair the damage to DNA and have an increased risk of BCC, SCC, and melanoma over time. BBCs can appear as shiny, waxy, pearly, red, or pink bumps, and SCC can appear as red, thick, scaly patches [63].

The management of patients with FA involves an annual whole-body skin examination and the use of sunscreen and sun-protective clothing.

### 3.5. Bloom Syndrome

Bloom syndrome (BS) is an autosomal recessive disease that is characterized by a strong predisposition to cancer and genetic instability. Only <300 cases have been reported in the Bloom Syndrome Registry [64]. It requires two pathogenic or likely pathogenic germline variants of the BLM gene, which encodes the DNA repair enzyme RecQL3 helicase. Patients present with dysmorphological features (a narrow face, a small lower jaw, and a prominent nose and ears) and pre- and postnatal growth retardation associated with erythema. The cancers that can develop in patients with BS are leukemia, lymphoma, and colorectal, larynx, breast, and skin cancers [65].

The principal characteristics of the skin, as seen in other DNA damage syndromes, are café-au-lait macules, but also poikiloderma and telangiectatic erythema of the face in a butterfly distribution, which can appear on the dorsum of the hands and forearms, and photosensitivity that may lead to cheilitis, crusting, bleeding, blistering, or erythema on any part of the body [65].

The skin cancers that can be seen in BS patients include BCC followed by SCC; they can appear on the head and neck as well as the arms, and are likely to represent an increased susceptibility to UV radiation [66].

The management of skin manifestations involves skin evaluations for rashes, abnormal nevi, or lesions that can be suspicious for BCC. Measures against the sun should be taken, such as sunscreen, sun-protective clothing, and decreased exposure to the sun [67].

### 3.6. Werner Syndrome

Werner syndrome (WS) is a genetic disorder characterized by a clinical feature suggestive of accelerated aging. It is caused by homozygous or compound heterozygous pathogenic or likely pathogenic variants of the WRN gene, with the role of resolving complex intermediate DNA structures [68]. WS has an incidence of 1 in a million newborns.

The skin manifestations are signs of aging, such as skin atrophy, the loss of subcutaneous fat, scleroderma-like changes, and the greying and loss of hair [68].

There are characteristic ulcers on the skin in approximately 40% of patients with WS, localized on the elbow joint, the knee joint, the distal one-third of the leg, and the medial malleoli [69].

Patients do not have the growth spurt, bilateral cataracts, or disorders that appear during middle age, such as type 2 diabetes mellitus, hypogonadism, osteoporosis, and atherosclerosis [68]. Patients with WS present with an increased incidence of sarcomas, but there are reports about thyroid follicular carcinomas, melanoma, meningioma, soft tissue sarcomas, and leukemia/myelodysplasia [68].

There is no specific treatment for the skin manifestations due to WS besides a careful examination, but the literature mentions the care of the ulcers when they appear with debridement to prevent infections and wound-healing measures such as negative pressure therapy or skin grafting [69].

### 3.7. Rothmund-Thomson Syndrome

Rothmund-Thomson syndrome (RTS) is an autosomal recessive genodermatosis, with a characteristic facial rash, heterogenous clinical features, and a predisposition to cancer. Only 300–400 cases have been reported. It is caused by homozygous or compound, heterozygous pathogenic, or likely pathogenic variants of the RECQL4 helicase gene [70].

The clinical heterogenous features include a short stature, sparse scalp hair, sparse or absent eyelashes and eyebrows, juvenile cataracts, skeletal abnormalities, radial ray defects, and premature aging [70]. The malignancies reported included osteosarcoma, SSC, BCC, and Bowe’s disease [70].

On the skin, the hallmark manifestation is a cutaneous rash with erythema and swelling and blistering of the face, which subsequently appears on the extremities and buttocks. Over time, the rash evolves to poikiloderma. Café-au-lait spots can also develop. Hyperkeratotic lesions and calcinosis cutis have also been documented [70].

An annual dermatological evaluation is recommended.

### 3.8. Dyskeratosis Congenita

Dyskeratosis congenita (DC) is a hereditary disease that is autosomal recessive, X- linked, or autosomal dominant and characterized by a triad that consists of reticulate hyperpigmentation, nail dystrophy, and oral leukoplakia. Its prevalence is 1 in a million newborns. DC is caused by germline pathogenic or likely pathogenic variants of 19 genes that directly or indirectly affect telomeres, with the most frequent being TERT, followed by RTEL1, CTC1, and DKC1. On the extracutaneous manifestations, patients can present with malignancies such as carcinoma of the gastrointestinal tract (hepatic, esophageal, pancreatic, rectum, or stomach), clear cell carcinoma, acute myeloid leukemia, Hodgkin’s lymphoma, myelodysplastic syndrome, or SCC of the larynx, tongue, and scrotum [71].

Reticulate hyperpigmentation (90%) on the skin could also be hypopigmentation, atrophy, or poikiloderma with telangiectasia. These lesions appear in sun-exposed areas, including the neck, trunk, and both extremities. Nail dystrophy (80%) is characterized by atrophy, longitudinal ridges, onychoschizia, pterygium, and anonychia. Premature greying, the loss of scalp hair and eyebrows, and sparse body hair are observed. Palmoplantar hyperkeratosis, chronic thumb ulceration, eczema, and hyperhidrosis have also been reported [71].

The management for dermatological manifestations is an exam of the skin and nails annually.

**Table 2 ijms-26-06140-t002:** Monogenetic diseases with cancer predisposition and skin manifestations.

Syndrome	Diagnostic Criteria	Dermatologic Manifestations	References
Xeroderma pigmentosum	Moriwaki et al. (2017) [72]	Photosensitivity, sunburning, skin pigmentation Skin cancers: SCC, BCC, melanoma, epithelioma, sebaceous cell carcinoma, fibrosarcoma, and angiosarcoma	[50,51]
Neurofibromatosis type 1	Kehrer-Sawatzki et al. (2022) [73]	CALMs, ephelides neurofibromas	[57]
FAMM	Soura et al. (2016) [58]	Numerous atypical nevi, melanoma	[58]
Fanconi anemia	Fanconi Anemia Clinical Care Guidelines (2020) [63]	CALMs, flexural hyperpigmentation, hypopigmented macules Skin cancers: BCC, SCC, and melanoma	[63]
Bloom syndrome	Langer 2006 [74]	CALMs, poikiloderma, telangiectatic erythema, photosensitivity Skin cancers: BSC, SCC	[66]
Werner syndrome	Oshima (2017) [68]	Atrophy of the skin, loss of subcutaneous fat, scleroderma-like changes, greying, loss of hair, ulcers on the skin	[68,69]
Rothmund-Thompson syndrome	Wang et al. [75]	Fibromas, epithelial cysts, desmoid tumors, lipomas, leiomyomas, neurofibromas, and pilomatricomas	[43]
Dyskeratosis congenita	AlSabbagh et al. [71]	Reticulate hyperpigmentation, hypopigmentation, atrophy of the skin, poikiloderma with telangiectasia, nail dystrophy, premature greying, loss of the hair and eyebrows, palmoplantar hyperkeratosis, chronic thumb ulceration, eczema, hyperhidrosis	[71]

## 4. Discussion

The clinical insights compiled in this review underscore the significance of dermatological manifestations as early indicators in various cancer predisposition syndromes (CPSs). These cutaneous signs may precede systemic symptoms and provide an accessible window for early suspicion and diagnosis. Recognizing the dermatological phenotype, therefore, plays a pivotal role in guiding patients toward an appropriate genetic evaluation and subsequent management strategies.

Dermatologists, in particular, are often the first specialists to encounter clinical clues suggestive of an underlying genetic syndrome, such as numerous melanocytic nevi in FAMM or multiple basal cell carcinomas in Gorlin syndrome. Similarly, the identification of atypical lesions, like BAP1-inactivated melanocytic tumors, requires nuanced clinical judgment and coordination with molecular diagnostics to guide patient management, even when the associated tumor type is not the presenting malignancy.

Moreover, dermatological surveillance has emerged as a formal recommendation in patients with known germline variants, such as BRCA1 and BRCA2 carriers, with an increased risk for melanoma. The involvement of dermatologists in routine follow-up facilitates the early recognition of secondary malignancies and provides an opportunity for a family-wide assessment through cascade testing.

Additional syndromes—for instance, Fanconi anemia, Bloom syndrome, and constitutional mismatch repair deficiency—present with dermatological signs that include café-au-lait macules, abnormal pigmentation, and telangiectasias. These phenotypes, when considered in conjunction with hematological or dysmorphic features, can trigger a timely referral to genetic services.

Notably, many CPSs present with hallmark skin lesions: trichilemmomas and oral papillomas in Cowden syndrome; desmoid tumors in familial adenomatous polyposis; or periorificial lentigines in Peutz–Jeghers syndrome. Other cutaneous tumors, such as sebaceous carcinomas or keratoacanthomas, may be the first clinical manifestation in syndromes like Muir–Torre. Therefore, establishing dermatological literacy across oncologic and genetic disciplines is vital for early identification and triage.

In the diagnostic landscape of hereditary cancers, interdisciplinary collaboration among dermatologists, oncologists, and geneticists is imperative. Dermatologists often serve as the first point of clinical detection, identifying suspicious cutaneous lesions through comprehensive skin examinations, often aided by dermoscopy. Once a lesion is identified, a biopsy is performed to evaluate its histological characteristics. Based on the clinical findings and family history, patients may be referred for a genetic consultation. Geneticists create a family tree, assess the cancer risk, and may recommend molecular testing to confirm a diagnosis. Following genetic confirmation, the care team jointly formulates a follow-up plan tailored to the specific syndrome, which may include clinical examinations, imaging studies, and cascade testing for at-risk family members. This coordinated, interdisciplinary workflow ensures an early diagnosis, proactive prevention, and personalized management.

When cutaneous findings are integrated with detailed family histories and personal oncological profiles, clinicians can better delineate at-risk individuals. This integrated, multidisciplinary model not only enhances the diagnostic accuracy, but also optimizes prevention, surveillance, and therapeutic intervention strategies for patients and their families.

## 5. Conclusions

Dermatological manifestations may serve as sentinel signs in patients with hereditary cancer syndromes. While some cutaneous features are common in the general population, their context within familial cancer patterns and syndromic associations mandates a comprehensive cancer risk assessment. Incorporating skin findings into genetic risk stratification can expedite the diagnosis and improve outcomes through personalized and precision medicine.

The management of CPSs must be inherently multidisciplinary. Dermatologists, geneticists, and oncologists each bring indispensable expertise that, when integrated, facilitates a precise approach to cancer diagnoses, treatment, and prevention. Understanding the dermatological phenotype allows not only for the timely treatment of skin lesions, but also for systemic cancer risk mitigation in affected individuals and their relatives.

## 6. Future Directions

Ongoing research should aim to elucidate the molecular mechanisms underpinning dermatological manifestations in CPSs, particularly through genotype–phenotype correlations.

Future clinical practice would benefit from consensus and international guidelines incorporating dermatologic signs into algorithms for hereditary cancers. Ultimately, developing formalized pathways that link dermatological evaluations with oncogenetic services will be essential in delivering equitable, precise, and proactive cancer care.

## Figures and Tables

**Figure 1 ijms-26-06140-f001:**
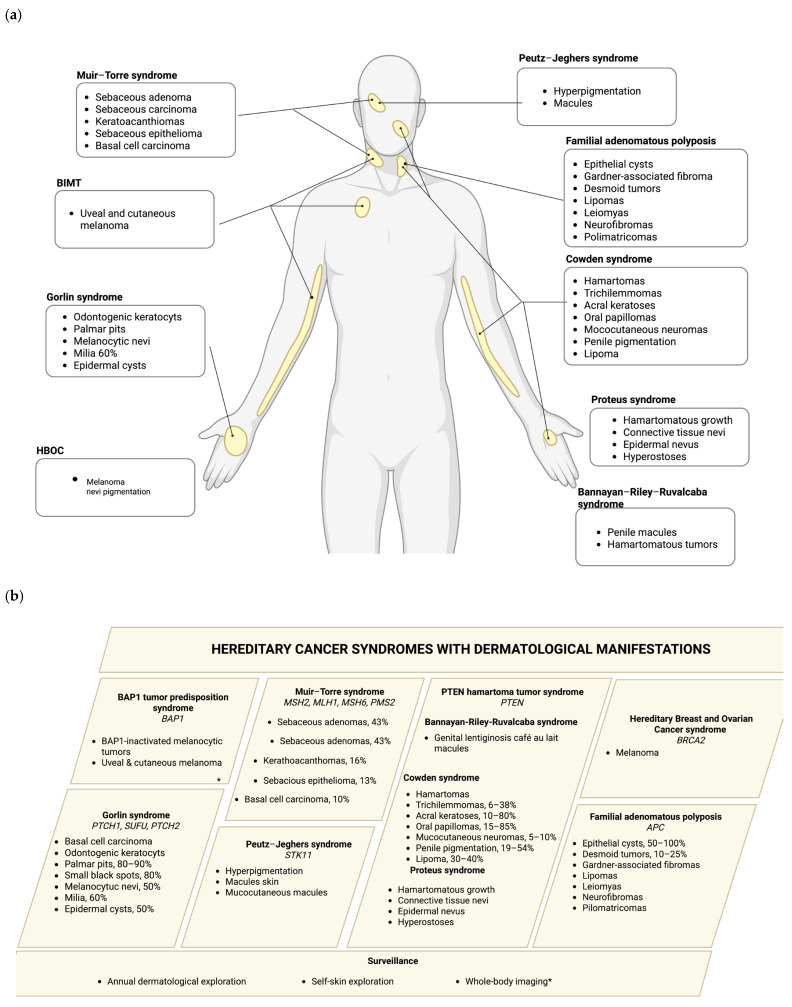
(**a**) Skin manifestations in hereditary cancer syndromes. Each square determines each of the hereditary cancer syndromes described with the dermatological manifestations, indicating the principal site of incidence and the frequency of the manifestation in each syndrome. (**b**) Recommended surveillance is also included; * indicates the surveillance specific to a particular syndrome. Created in biorender.com.

**Figure 2 ijms-26-06140-f002:**
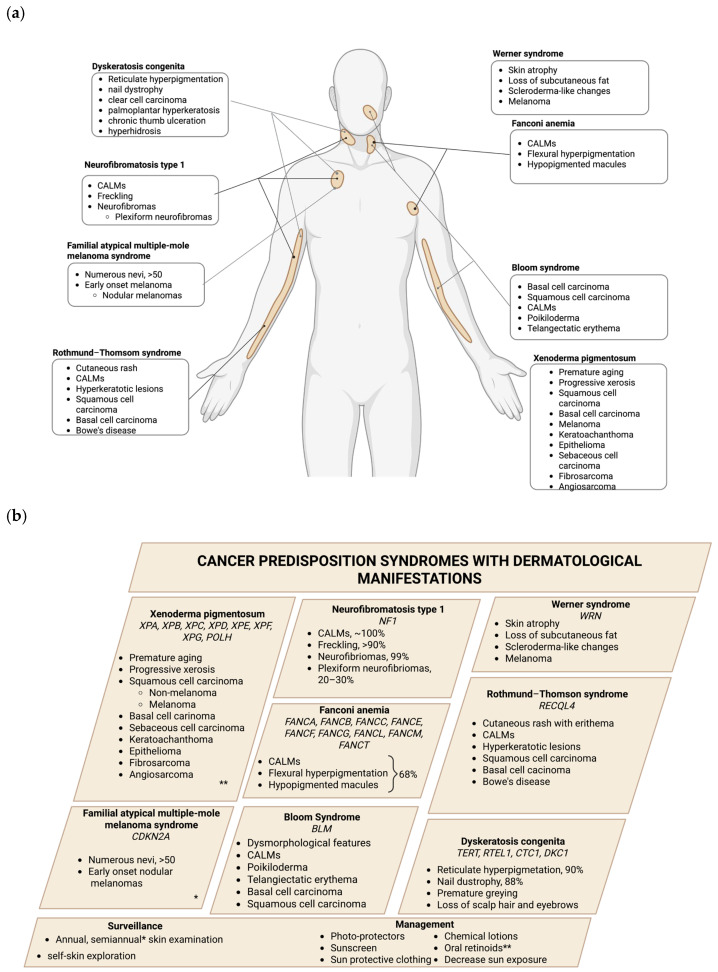
Monogenetic diseases with cancer predisposition and skin manifestations. (**a**) Each square determines each of the monogenetic diseases and the principal skin manifestations, while each line determines the principal sites of incidence of the skin manifestations and the frequency of the manifestation. (**b**) Recommended surveillance and management are also included, where * indicates surveillance and ** indicates management specific to a syndrome. Created in biorender.com.

## Data Availability

No new data were created or analyzed in this study. Data sharing is not applicable to this article.

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
