# Peer review of "Skin Signals: Exploring the Intersection of Cancer Predisposition Syndromes and Dermatological Manifestations"

_ijms, 2025, doi:10.3390/ijms26136140_

Round 1
Reviewer 1 Report
Comments and Suggestions for Authors
The aim of this review is to provide a comprehensive overview of dermatological manifestations in hereditary cancer predisposition syndromes, emphasizing the diagnostic significance of cutaneous signs. As a result of bridging dermatology and cancer genetics, the manuscript is highly relevant, but major revisions are required before it can be published. Particularly, authors should improve the aspects detailed below:
- Redundancy and Organization: Several concepts are repeated across sections (e.g., cutaneous signs as early indicators of syndromes), which could be consolidated for conciseness. The overall structure would benefit from clearer section headings and smoother transitions between syndromes.
- Scientific Rigor: some associations (e.g., between BRCA2 and melanoma) are presented without proper context or are still under debate. In such cases, references to consensus guidelines or systematic reviews should be included.
- Tables and Figures: the summary tables are valuable but dense and, at times, difficult to read. Improving formatting and editing for grammar and consistency is necessary.
- Introduction: clarify the scope of the review and its target audience.
- Discussion and Conclusions: the discussion lacks depth in critical analysis and synthesis of the information presented. It could better emphasize how dermatologists, geneticists, and oncologists can collaborate.
- Terminology: introduce acronyms clearly at first use (e.g., LS, MTS, XP). Ensure consistent use of terms (e.g., use either "melanocytic tumors" or "melanocytic neoplasms" throughout).
It is highly recommended that thorough professional English editing be conducted to improve clarity and consistency. Many sentences are poorly structured, redundant, or confusing.
Reviewer 2 Report
Comments and Suggestions for Authors
The submitted manuscript entitled "Skin Signals: Exploring The Intersection Of Cancer Predisposition Syndromes And Dermatological Manifestations," is a comprehensive review that effectively addresses its stated aim. It provides a detailed overview of various hereditary cancer predisposition syndromes and their associated dermatological manifestations, emphasizing the importance of early identification for improved patient surveillance. The writing is generally clear and descriptive, making complex medical information accessible. The language is appropriate for a scientific review, targeting dermatologists, oncologists, and medical geneticists. While mostly well-written, there are minor instances of awkward phrasing and grammatical errors that could be polished for enhanced flow and professionalism. It fully delivers on this aim, covering a wide array of syndromes. This topic is highly relevant for clinicians involved in cancer diagnosis and management, as early recognition of skin signs can significantly impact patient outcomes. The depth of detail is a significant strength for a review article. Yet, The section on "Xeroderma Pigmentosum" (Section 3.1) abruptly cuts off. This needs to be completed to maintain the comprehensiveness of the review. Also, The numbering in Section 3 restarts at "1.1" instead of continuing sequentially (e.g., "3.1"). This is a minor point but should be corrected for consistency.
Round 2
Reviewer 1 Report
Comments and Suggestions for Authors
The revised version of the manuscript demonstrates substantial improvements in content, clarity, and structure, and it addresses the core of the requested revisions. The manuscript could be accepted for publication after minor revisions:
-The revised version reduces redundancy, sections are more structured with clearer headings and smoother transitions. A better effort is made to group syndromes logically. However some repetition persists (e.g., reiterating the role of dermatologists in early detection in multiple sections).
-Tables are still dense and would benefit from visual simplification (e.g., grouping by syndrome type).
-Minor grammar inconsistencies remain (e.g., “Dermathologic” should be “Dermatologic”).
-The new discussion section better emphasizes interdisciplinary collaboration, however it could better explore clinical implications or propose a workflow for integrating dermatologic findings in cancer genetics.
